# Immersive Virtual Reality Environments as Psychoanalytic Settings: A Conceptual Framework for Modeling Unconscious Processes Through IoT-Based Bioengineering Interfaces

**DOI:** 10.3390/bioengineering12111257

**Published:** 2025-11-17

**Authors:** Vincenzo Maria Romeo

**Affiliations:** 1Department of Culture and Society, University of Palermo, Viale Delle Scienze, Ed. 15, 90128 Palermo, Italy; vincenzomaria.romeo@unipa.it; Tel.: +39-3405803854; 2School of Psychoanalytic and Groupanalytic Psychotherapy S.P.P.G., Via Fontana n° 1, 89131 Reggio Calabria, Italy; 3Neurosinc, Via Paolo Bentivoglio n° 62, 95125 Catania, Italy

**Keywords:** psychoanalysis, immersive virtual reality (IVR), internet of things (IoT), affective computing, heart rate variability (HRV), galvanic skin response (GSR), EEG, symbolization, transference, rêverie

## Abstract

**Background:** Immersive Virtual Reality (IVR) is gaining increasing relevance in the field of mental health as a tool for therapeutic simulation and embodied experience. However, most existing VR applications are grounded in cognitive–behavioral frameworks, leaving unexplored the integration of symbolic, intersubjective, and unconscious dimensions. Psychoanalysis—particularly its constructs of setting, rêverie, and transference—offers a unique epistemological basis for designing therapeutic environments that engage implicit emotional processes. **Aim:** This paper aims to develop a conceptual framework for modeling IVR-based therapeutic settings inspired by psychoanalytic theory and enhanced through IoT-enabled biosensing technologies. **Methods/Approach:** We propose a three-layer architecture: (1) a somatic layer involving IoT-based real-time physiological monitoring (e.g., heart rate variability, galvanic skin response, eye-tracking, EEG); (2) a symbolic-narrative layer where the VR environment dynamically adapts to the user’s affective state through immersive visual and auditory stimuli; and (3) a relational layer where AI-driven avatars simulate transferential dynamics. The model is theoretically grounded in psychoanalytic literature and informed by current advances in affective computing and bioengineering. **Conclusions:** By bridging psychoanalytic metapsychology and bioengineering design, this framework proposes a novel approach to therapeutic IVR systems that move beyond explicit cognition to engage the embodied unconscious. The integration of IoT biosignals enables the mapping and modulation of internal states within a structured symbolic space, opening new pathways for the clinical application of digital psychoanalysis.

## 1. Introduction

The integration of immersive virtual reality (IVR) into clinical and therapeutic contexts has gained considerable momentum in recent years, driven by the exponential development of biosensors, wearable devices, and affective computing systems. IVR environments are now being employed across diverse domains, from exposure therapy for phobias and PTSD to neurorehabilitation and pain management, offering patients the possibility to engage with simulated yet affectively potent scenarios in a controlled, customizable manner [1,2]. Within the field of mental health, this technological shift has opened new pathways for embodied simulation and experiential engagement, facilitating a kind of “prosthetic imagination” capable of reconfiguring perception, cognition, and affect [3].

However, while the psychophysiological foundations of VR-based intervention have been extensively explored in cognitive and behavioral paradigms, the symbolic, unconscious, and relational dimensions of subjectivity remain largely underrepresented. Current frameworks tend to focus on explicit behaviors, rational decision-making, and observable symptoms, often overlooking the deeper layers of subjective meaning, affect regulation, and transference dynamics that structure psychological life [4,5]. From a clinical perspective, this gap is particularly significant in cases where patients exhibit dissociative defenses, trauma-related symbolic disorganization, or identity diffusion, which are not adequately addressed by surface-level interventions [6].

Psychoanalysis, with its emphasis on the unconscious, symbolic elaboration, and the intersubjective matrix of mental functioning, provides a rich and underutilized epistemological foundation for informing the design of IVR environments. Constructs such as the setting, understood not merely as a physical space but as a psychic frame that contains and organizes experience [7], or rêverie, conceptualized by Bion as the analyst’s capacity to metabolize the patient’s unconscious emotional communication [8], have critical potential in modeling therapeutic immersion. Moreover, the dynamics of transference and countertransference, core to analytic process, find novel and urgent relevance in virtual environments, particularly when avatar-based relational interfaces are employed [9].

The advent of Internet of Things (IoT)-enabled biosensors—including heart rate variability (HRV), galvanic skin response (GSR), eye-tracking, EEG, and respiratory monitoring—introduces the opportunity to detect and respond to somatic correlates of affective states in real time [10,11]. This technological capability opens a space for integrating embodied affective data into symbolic-narrative constructions, whereby the virtual environment adapts not just to explicit user actions, but to implicit physiological signals corresponding to unconscious emotional processes. The resulting architecture allows for the generation of responsive symbolic landscapes, which mirror the user’s internal world and provide a platform for psychic transformation through symbolic re-elaboration.

Despite the promise of this approach, a coherent and theoretically grounded model for integrating psychoanalytic constructs with immersive technology and IoT-based bioengineering systems remains lacking in the literature. Most IVR frameworks continue to rely on linear stimulus-response models, without incorporating the recursive, associative, and deeply embodied logic that underpins psychodynamic functioning [12]. This absence not only limits the clinical efficacy of current digital tools but also reflects a broader epistemological disconnect between psychodynamic theory and technological design.

The purpose of this paper is to develop a conceptual framework for immersive virtual environments informed by psychoanalytic theory and supported by IoT-enabled biosensing technologies. We propose a three-layer architecture: (1) a somatic layer, composed of physiological monitoring systems capable of detecting affective states; (2) a symbolic-narrative layer, where the virtual setting responds adaptively to internal signals by modulating environmental features and narrative content; and (3) a relational layer, in which avatars and AI-driven agents are designed to simulate or engage with transferential dynamics. This framework aims to bridge the gap between symbolic-affective depth and technological infrastructure, opening new clinical and theoretical frontiers for therapeutic VR systems.

### Related Work (2020–2024)

Recent syntheses have consolidated the clinical evidence for virtual reality psychotherapy across anxiety disorders, PTSD, and other conditions, emphasizing both effectiveness and translational considerations for routine care [13,14,15]. Within this landscape, the present work extends beyond exposure-centric models by specifying a closed-loop design that couples biosignal-driven affect estimation with symbolic environmental modulation and relational attunement—thus addressing dimensions of meaning-making typically underrepresented in mainstream VR protocols. In parallel, rapid advances in artificial intelligence highlight opportunities and risks of large language models (LLMs) and conversational agents for mental-health support. Recent overviews and meta-analyses document that AI-based conversational systems can support psychoeducation and symptom reduction, and can even establish user-perceived working alliance under specific design and safety constraints [16,17]. At the same time, position papers and roadmaps stress the high-stakes nature of psychotherapy, the necessity of guardrails, and the limits of current LLMs for clinical interpretation [18,19]. Against this background, our avatars are explicitly non-interpretive: they do not deliver clinical judgments; rather, they mirror and attune to biosignal-derived affect states through constrained symbolic adjustments (light, space, sound, transitional objects) with on-device processing where feasible and opt-in for optional modules (e.g., prosody), preserving privacy and clinician oversight.

## 2. Psychoanalytic Framework

While immersive virtual reality (IVR) and biofeedback technologies are advancing rapidly in therapeutic contexts, there remains a notable absence of models integrating unconscious processes, symbolic meaning, and affective depth into system design. Psychoanalysis, with its complex metapsychology and enduring clinical relevance, offers a sophisticated conceptual framework through which therapeutic technologies can be reconceptualized—not only as tools for cognitive change, but as environments for transformation rooted in the dynamics of the unconscious [20,21] (see Box 1, Glossary).

Box 1**Glossary**.**Analytic third**. The co-created intersubjective field that emerges between patient and clinician; neither party’s mind alone, but a relational matrix that shapes meaning, affect regulation, and interpretation.**Rêverie**. The clinician’s receptive state of mind that absorbs and metabolizes unprocessed affect into thinkable experience; in this IVR model, the environment simulates a rêverie-like function via gentle, symbolically resonant adaptations.**Symbolic modulation**. The real-time, non-interpretive adjustment of VR parameters (light, spatial density, soundscape, transitional objects) driven by biosignal/behavioral cues to foster symbolization and affect regulation under safety-first rules.**Holding**. The reliable, protective frame that contains anxiety and supports psychic organization; in IVR it includes a neutral safety scene, predictable transitions, panic-exit, and conservative defaults when arousal is high.**Transitional object/space**. A mediating artifact/zone between inner and outer reality that supports play and integration; in IVR it appears as interactable symbolic objects and liminal areas (e.g., thresholds, corridors) that invite projection and meaning-making.

### 2.1. The Analytic Setting as a Holding Structure

The analytic setting has long been theorized not merely as a physical space, but as a psychic frame that maintains the structural conditions necessary for emotional containment and symbolic elaboration. José Bleger was among the first to emphasize the importance of the setting as an “encapsulated mute framework”, a kind of background object that, when disrupted, exposes archaic aspects of the self [22]. Winnicott further developed the notion of the “holding environment”, conceptualizing it as the psychophysical context in which the patient’s primitive anxieties can be metabolized through the constancy and attunement of the analyst [23]. Translated into digital environments, this suggests that virtual spaces must not only provide interactional affordances, but also replicate the function of psychic containment, allowing for the surfacing and modulation of unconscious content.

### 2.2. Rêverie and the Maternal Function

The construct of rêverie, as introduced by Wilfred Bion, defines the analyst’s capacity to receive, absorb, and transform the patient’s raw emotional projections—what he termed “beta-elements”—into more tolerable, symbolically processed forms [24]. Rêverie is thus not a passive fantasy, but an active emotional processing function that enables the transformation of unmentalized states. In a technological setting, particularly one equipped with IoT-based affective sensors, rêverie could be algorithmically modeled through the adaptation of visual, auditory, or spatial elements in response to physiological markers (e.g., changes in heart rate variability, GSR, EEG). This process would render the environment capable of symbolic resonance, offering the user a form of “digital rêverie” that reflects and processes their emotional states in real time.

Moreover, rêverie represents an essential function of non-verbal maternal responsiveness, one that could be simulated through ambient changes in the immersive VR environment (e.g., light dimming, texture modulation, auditory containment). These shifts would not be reactive in a behavioral sense, but transformative—aesthetic and affective modifications that respond to unconscious states and support symbolic elaboration [25].

### 2.3. Transference and the Symbolic Order

Transference is the process by which the patient projects internal object relations onto the therapist, re-enacting relational patterns from early developmental experiences. In analytic terms, transference is not an error to be corrected, but a repetition to be interpreted and symbolized [26]. In an IVR environment, particularly one where avatars or agents are deployed, transferential dynamics may emerge in nuanced and unexpected ways. For instance, a virtual assistant who mimics vocal prosody or proximity patterns may unconsciously activate relational memories, affective scripts, or defensive operations.

From a system design perspective, the potential for relational encoding within avatars or affective agents could be deliberately employed to simulate and reflect transferential configurations. The patient’s physiological data—e.g., increased arousal during proximity or eye contact with an avatar—could be interpreted as a transferential signal, prompting the system to either intensify or soften the interaction, depending on the desired therapeutic trajectory [27].

Furthermore, Jacques Lacan’s concept of the Symbolic Order—the structured system of language, law, and meaning—invites reflection on the architecture of the digital environment itself as a mediating symbolic frame. In VR, every object, movement, and spatial parameter constitutes a signifier within a symbolic network, capable of embodying aspects of the unconscious. The way a user navigates these signifiers, responds to their affordances, or resists their logic may offer important insights into their psychic structure [28].

### 2.4. The Transitional Space and the Use of the Object

Another key psychoanalytic construct that is directly applicable to immersive VR environments is Winnicott’s notion of the transitional space—a liminal zone between internal and external reality in which creativity and symbolic play emerge [29]. In this space, the subject is neither fully merged with the object nor completely separate, allowing for the development of symbolic autonomy and the capacity to use objects as carriers of meaning rather than as extensions of the self.

Immersive VR environments, if properly structured, can serve as technologically-mediated transitional spaces, enabling patients to experiment with representations of the self, explore relational dynamics, and project internal states onto external virtual structures. This potential is magnified when the VR setting includes transitional objects—virtual elements that carry subjective meaning and can be manipulated, abandoned, or transformed by the user [30].

In this sense, the therapeutic potential of VR does not lie solely in exposure or training, but in the symbolic reorganization of the subject’s internal world. Such reorganization depends not on behavioral reinforcement, but on the capacity of the environment to mirror, contain, and transform unconscious material—functions that are at the heart of the psychoanalytic process.

### 2.5. From Intersubjectivity to Technologically Mediated Embodiment

Modern relational and intersubjective psychoanalysis has emphasized that the psyche is not isolated, but co-constructed in dynamic relationships with others. Within VR environments, this opens the door to technologically-mediated intersubjectivity, where avatars, voice feedback, and environmental responsiveness create a simulated relational field that can be therapeutically engaged [31].

Crucially, this field must not be overdetermined or prescriptive. As in analytic therapy, uncertainty, ambiguity, and open-endedness are essential conditions for symbolic exploration. A VR setting too tightly scripted or directive risks becoming an intrusive superego, rather than a facilitating environment. Therefore, the integration of psychoanalytic principles into IVR must preserve the space for psychic emergence, allowing meaning to be co-created, rather than imposed [32].

## 3. Bioengineering and IoT-Based Interfaces

The convergence of bioengineering, biosensing technologies, and immersive virtual reality (IVR) environments has generated new possibilities for real-time interaction between physiological signals and dynamically adaptive digital systems. In therapeutic and neurocognitive domains, this integration—often conceptualized under the umbrella of the Internet of Things (IoT)—enables the collection, processing, and feedback of biometric data for both research and clinical purposes [33]. Within a psychoanalytic IVR framework, such interfaces may serve not merely as passive sensors but as active mediators of unconscious communication, linking the embodied dimension of affect to symbolic modulation in virtual space.

### 3.1. The Role of Biosensors in Affective Computing

IoT-based biosensors are engineered to detect a range of physiological parameters with high temporal resolution, enabling the construction of affective computing platforms capable of inferring users’ emotional and cognitive states [34]. The field is built upon the assumption that autonomic nervous system responses—such as heart rate variability (HRV), galvanic skin response (GSR), electroencephalographic (EEG) activity, and eye movement dynamics—can serve as reliable indicators of psychophysiological arousal, attention, and stress [35].

In immersive VR environments, these biometric streams can be integrated in real time through closed-loop feedback systems, whereby changes in the user’s emotional state trigger transformations in the environment’s architecture—altering lighting, spatial configuration, narrative elements, or avatar behavior. Such responsiveness, when conceptualized psychoanalytically, becomes a form of embodied rêverie, wherein the system reflects and reconfigures unconscious content expressed through the body [36].

Importantly, biosensors employed in these contexts must ensure minimal intrusiveness, high fidelity, and stable wireless connectivity, particularly in full-body motion scenarios. Wearable devices such as chest straps, smartwatches, dry EEG headbands, and skin-mounted electrodes are commonly utilized in research-grade and clinical-grade systems [37,38].

### 3.2. Core Physiological Parameters for Unconscious Affect Mapping

To create an analytic IVR environment capable of mirroring unconscious processes, it is essential to identify the biosignals most relevant for tracking affective states and implicit relational dynamics. The following parameters are particularly pertinent:Heart Rate Variability (HRV): HRV reflects parasympathetic nervous system tone and is a well-established index of emotional regulation, interoceptive awareness, and resilience [39]. In psychoanalytically-informed settings, fluctuations in HRV may correspond to moments of affective activation, anxiety, or psychic withdrawal, particularly in response to symbolic stimuli.Galvanic Skin Response (GSR): Also referred to as electrodermal activity (EDA), GSR captures skin conductance changes due to sweat gland activity. It is a sensitive index of sympathetic arousal and has been shown to correlate with unconscious anxiety, attentional shifts, and affective memory retrieval [40].Electroencephalography (EEG): Non-invasive EEG allows monitoring of neural oscillations related to attention, memory, and emotional engagement. In therapeutic VR, increased frontal theta or alpha asymmetry may indicate shifts in affective regulation and mentalizing capacities, both of which are central to analytic work [41].Eye-Tracking: Gaze fixation duration, pupil diameter, and saccadic movement are powerful proxies for interest, aversion, and emotional resonance [42]. In psychoanalytic terms, eye-tracking could be used to detect unconscious avoidance or hyper-investment in certain symbolic configurations or avatars, thus informing adaptive changes in the virtual environment.Respiration Rate and Depth: Respiratory dynamics are intimately linked to emotional states and defensive postures. Shifts toward shallow breathing or breath-holding may indicate repression, inhibition, or defensive withdrawal—phenomena particularly salient in trauma-related conditions [43].

Together, these physiological indices can form a multi-modal affective profile of the user, updated in real time and processed by a supervisory algorithm that modulates the symbolic features of the VR environment accordingly (see Table 1 for a concise mapping between IoT biosignals, psychoanalytic interpretations, and environmental symbolic modulation).

Quantitative ranges and trigger rules are reported in Appendix A.

### 3.3. IoT System Architecture in Analytic-VR Environments

The IoT system designed for a psychoanalytic IVR architecture must integrate sensor inputs, data processing units, and dynamic content rendering. A typical structure includes:Sensor Nodes: Wearable or ambient biosensors wirelessly transmit raw data to a central processing unit. These include ECG modules (for HRV), EDA sensors (for GSR), EEG headsets, and infrared eye-trackers. Modern systems often adopt Bluetooth Low Energy (BLE) or Wi-Fi communication protocols for real-time streaming [44].Edge Computing Layer: Data preprocessing and artifact removal are performed at the edge (on-device or near-device computing), reducing latency and preserving privacy. Algorithms here handle signal normalization, peak detection, and transformation into emotional markers [45].VR Rendering Engine Integration: The processed affective states are mapped onto VR content parameters using emotion-adaptive algorithms. For instance, high sympathetic arousal may trigger the dimming of lights, softening of textures, or activation of soothing soundscapes in the environment—functions that replicate the psychic “holding” seen in human rêverie [46].Feedback Loop and Logging: A feedback loop allows continuous adjustment of VR parameters in response to real-time biometric changes. All data are logged and time-stamped, enabling post-session clinical review and psychodynamic interpretation.

This architecture aims not just to optimize user engagement, but to construct a technologically-mediated analytic frame, one that re-embodies the symbolic function of the therapeutic encounter.

### 3.4. Proof-of-Concept and Validation Roadmap

Scope. This study advances a conceptual framework; we therefore outline a staged plan to test feasibility, safety, and face validity.Minimal setup. Unity/Unreal engine with a BLE/Wi-Fi listener; HRV (RMSSD/SDNN), EDA (phasic peaks/tonic SCL), eye-tracking (fixation/avoidance), optional respiration belt; closed-loop update at 5–10 Hz.Technical benchmarks. Streaming latency < 100 ms; packet-loss < 2%; stable scene parameter updates without frame-rate drops.Feasibility POC. N = 10–15 healthy adults; 2–3 short sessions (10–12′). Outcomes: presence (IPQ/ITC-SOPI), cybersickness (SSQ), usability (SUS/UEQ), trends in autonomic regulation (↑RMSSD, ↓EDA peaks) during symbolic “holding” phases. Stop-rules for distress and a neutral safety scene are enforced.Pilot extension. In a psychodynamic clinic sample (N ≈ 25–30), assess acceptability, safety, and exploratory clinical signals; refine mapping thresholds and privacy safeguards.Data handling. On-device preprocessing, minimal retention, de-identification; optional voice/prosody disabled by default unless explicit opt-in.

### 3.5. Behavioral and Interaction Metrics

Beyond physiological streams (HRV/EDA/EEG/eye-tracking), we include a behavioral channel capturing head/hand kinematics, locomotion (approach–avoidance), postural expansion–constriction, object/-avatar interaction logs, and optional vocal prosody (F0 variability, speech rate, pause ratio). These features complement implicit autonomic indices and are fused using a safety-first multimodal scheme (early fusion for time-locked micro-events; late fusion for scene-level decisions). When physiological arousal is high and behavioral avoidance is detected, the system privileges containment (e.g., dimmer light, lower spatial density, avatar distance). Voice is optional and processed on-device; no raw audio is stored.

### 3.6. Ethical and Technical Considerations

While the integration of biosensors into therapeutic environments holds great promise, it raises important ethical and technical concerns. Data privacy, emotional safety, and algorithmic transparency are critical elements in the design of such systems. Users must be informed about the nature of biometric monitoring, the emotional implications of adaptive environments, and the limits of interpretation [47].

Moreover, from a psychoanalytic standpoint, the risk of technological overdetermination—whereby symbolic uncertainty is supplanted by interpretive closure—must be carefully mitigated. The aim is not to decode the unconscious in mechanistic terms, but to facilitate its symbolic elaboration through embodied, affective engagement with the VR environment.

## 4. Conceptual Model: Psychoanalytic IVR Architecture

This section presents the core architecture of a conceptual framework designed to integrate psychoanalytic principles with immersive virtual reality (IVR) systems and IoT-enabled biofeedback technologies. The model proposes a three-layer structure that reflects the interdependence of embodied affectivity, symbolic elaboration, and relational dynamics—key dimensions in psychoanalytic theory and therapeutic practice. By translating these clinical constructs into technological domains, we propose a bio-symbolic analytic setting, which can dynamically adapt to the user’s implicit signals and facilitate psychic transformation through immersive engagement.

### 4.1. Overview of the Three-Layer Framework

The proposed architecture consists of the following components:Layer 1—Somatic–Sensorial Input Layer: Real-time detection and interpretation of physiological and behavioral data via IoT biosensors.Layer 2—Symbolic–Narrative Modulation Layer: Adaptive modification of environmental features and symbolic representations within the IVR environment based on input from Layer 1.Layer 3—Relational Interface Layer: Dynamic relational simulation via avatars or agents that can engage in transferential dialogues informed by user responses and psychophysiological profiles.

Each layer communicates bi-directionally, allowing for recursive feedback, symbolic re-elaboration, and the construction of a personalized, affectively attuned virtual environment (Figure 1).

The Input Layer (bottom) collects real-time biosignals (e.g., heart rate variability, GSR, EEG) via IoT-based wearable interfaces.The Symbolic Modulation Layer (middle) translates affective and somatic data into environmental features (e.g., lighting, space, avatar expression), functioning as a dynamic rêverie system.The Clinical Integration Layer (top) aligns the immersive experience with psychoanalytic constructs such as transference, containment, and narrative coherence, supporting unconscious elaboration and therapeutic symbolization. The model aims to provide a scalable, adaptive infrastructure for psychodynamically oriented digital health interventions.

### 4.2. Layer 1: Somatic-Sensorial Input Layer

This foundational layer integrates biometric signals collected via wearable or ambient IoT devices. Parameters include:Heart rate variability (HRV), reflecting parasympathetic activity and emotional regulationGalvanic skin response (GSR), indexing sympathetic arousalEEG markers (e.g., frontal alpha asymmetry), associated with emotional valence and attentionEye-tracking, measuring gaze direction, fixation, and pupil dilationRespiration and motion data, indicating relaxation, arousal, or freeze responses

These inputs are processed through an affective recognition algorithm that generates a dynamic map of the user’s internal state. Rather than producing explicit diagnostic outputs, the goal is to generate a latent affective profile, akin to Bion’s “beta-elements” awaiting symbolic transformation [48].

Within a psychoanalytic logic, this layer functions as a sensorial unconscious, providing the raw material from which deeper symbolic and relational responses emerge. The data are then transferred to Layer 2 for affect-responsive modulation. See Figure 2 for the technical architecture and I/O flows.

A high-resolution, numbered version is provided in Appendix A.

### 4.3. Layer 2: Symbolic-Narrative Modulation Layer

The second layer involves the adaptive transformation of the IVR environment in response to the biometric signals received. This modulation occurs on three primary levels:Environmental Aesthetics: Light intensity, color temperature, visual textures, and ambient soundscapes are adjusted to reflect and contain the user’s emotional state. For instance, elevated arousal might trigger softening of light and sound—an environmental rêverie aiming to metabolize unprocessed affect [49].Spatial Dynamics: The spatial organization of the environment adapts based on physiological indicators of withdrawal or engagement. Expansive spaces may close in during dissociative states, while constricted paths may open when users demonstrate readiness to explore. This symbolic shifting mirrors the subject’s internal relational field [50].Narrative and Symbolic Elements: Archetypal imagery, interactive objects, and dreamlike sequences may emerge responsively. For example, a transitional object (e.g., a floating cube) may become more prominent when GSR indicates emotional intensity, inviting projection and symbolic manipulation. These objects are not pre-determined but emergent within the user’s affective-symbolic space, allowing for intersubjective meaning-making [51].

This layer thereby embodies the function of rêverie, understood not merely as reflection, but as transformation: a mediating symbolic matrix that absorbs, represents, and reshapes the user’s unconscious emotional states in a way that facilitates psychic integration.

### 4.4. Layer 3: Relational Interface Layer

This layer introduces AI-driven avatars or semi-autonomous agents that simulate relational dynamics within the IVR setting. These agents can interact through speech, gesture, gaze, and proximity—modalities which, when modulated, evoke different transferential configurations.

Affective Synchrony and Rupture: The avatar can increase or decrease engagement depending on biometric feedback. For instance, when HRV indicates hyperactivation, the avatar may take a more distant posture or adopt a slower vocal rhythm. Such shifts are not behavioral adaptations alone, but simulations of analytic attunement [52].Simulation of Transference: The avatar’s role is not simply supportive; it may take on transferential positions (e.g., idealized, persecutory, maternal) based on user interactions and affective patterns. This does not replace the analyst but models transferential emergence, allowing the patient to experiment with relational scripts in a transitional space [53].Intersubjective Resonance: Advanced models of affective computing have shown that emotional synchrony between human and avatar enhances therapeutic alliance and emotional depth [54]. In this architecture, the avatar operates within ethical boundaries: not interpreting or diagnosing, but responding with calibrated emotional mirroring that facilitates symbolization.

The avatar does not replace the human clinician but offers a mediated simulation of intersubjectivity, preparing the ground for deeper therapeutic work either in virtual or hybrid analytic formats.

### 4.5. Feedback Loops and Recursive Adaptation

The three-layer model is structured as a recursive system: physiological data (Layer 1) modulate symbolic parameters (Layer 2), which in turn influence relational behaviors (Layer 3), which then elicit new physiological and behavioral responses from the user, restarting the loop. Algorithm 1 summarizes the closed-loop control logic.
**Algorithm 1** Multimodal Affective-to-Symbolic Control (Closed Loop)Inputs:Physio P_t = {HRV_RMSSD, EDA_phasic, EDA_tonic, EEG_alpha_asym, EyeFix}Behavior B_t = {head/hand kinematics, locomotion (approach–avoidance),object/avatar interactions, (prosody: F0, rate, pauses) [optional]}Parameters:θ_vagal, θ_arousal        # safety thresholdsf_s = 5–10 Hz                # control-loop frequencyw_phys, w_beh              # fusion weights (safety-first priority)Initialize:scene_params ← neutral_safe_sceneavatar_params ← neutral_presencewhile session_active:# 1) Acquisition & preprocessing (edge)P̂_t ← preprocess(P_t)         # filtering, artifact removal, z-scoringB̂_t ← preprocess(B_t)         # smoothing, normalization# 2) Affect proxies (physio & behavior)p_affect ← g_phys(P̂_t)        # e.g., vagal withdrawal (HRV), arousal (EDA), avoidance (EyeFix)b_affect ← g_beh(B̂_t)          # e.g., approach–avoidance, freezing, interaction tempo# 3) Safety-first multimodal fusionaffect* ← fuse(p_affect, b_affect; weights = {w_phys, w_beh}, safety_priority = True)# 4) Symbolic/environmental mappingscene_params ← map_env(affect*)        # LightIntensity, SpaceDensity, Soundscape, TransitionalObjects# 5) Relational/Avatar adaptationavatar_params ← map_avatar(affect*)    # Proximity, Gaze, Turn-taking/Prosody# 6) Render & logrender(scene_params, avatar_params)log(t, P̂_t, B̂_t, p_affect, b_affect, affect*, scene_params, avatar_params)# 7) Safety overrideif (p_affect.arousal > θ_arousal) or (b_affect.avoidance_high) or user_panic_exit():scene_params ← neutral_safe_sceneavatar_params ← calming_distancerender(scene_params, avatar_params)continueend while

In this architecture, HRV_RMSSD is used as a proxy for vagal tone, the phasic component of electrodermal activity (EDA_phasic) indexes sympathetic arousal, and EyeFix denotes ocular fixation/avoidance patterns. Multimodal fusion follows a safety-first policy: when inferred arousal is high, containment actions take precedence over exploration-driven adaptations. The closed-loop controller operates at 5–10 Hz. All raw audio input is optional, processed on-device only, and never stored. This circular causality reflects the dynamic unconscious, where meaning is continuously reshaped by experience and symbolization.

This architecture, when visualized (see Figure 1), mirrors the triangular structure of the analytic field: the body (sensorial), the symbolic (setting/rêverie), and the relational (transference). Each layer is essential, and none can be reduced to the others. Their integration forms a transitional digital space where emotional truths can be encountered, represented, and potentially transformed. For implementation-level details, see Appendix A.

### 4.6. Clinical Utility and Personalization

The model is designed to accommodate personalized therapeutic trajectories:For trauma survivors, the environment can be tuned to detect physiological flashbacks and modulate sensory inputs accordingly, preventing retraumatization.In borderline structures, the relational layer may simulate stable presence and affective reliability, modulated by real-time signals of abandonment anxiety.In patients with psychosomatic symptoms, the system may track and mirror non-verbal expressions of distress through symbolic sequences, fostering embodied insight [48].

By interweaving biosignals, symbolic logic, and relational dynamics, this model proposes a new paradigm of embodied digital therapy, informed by the epistemology of psychoanalysis and operationalized through advanced bioengineering technologies.

The full controller specification is detailed in Appendix A.

## 5. Symbolic Environmental Design

While biosensors and avatar interfaces provide the technological scaffolding for a psychoanalytically informed IVR system, the symbolic design of the immersive environment remains central to enabling unconscious elaboration. In psychoanalytic theory, the capacity to symbolize—to represent affective experience through metaphoric, imaginal, or spatial transformation—is a core function of psychic health [55]. Consequently, the construction of symbolic virtual environments must go beyond aesthetic representation to become containers of meaning, where subjective experience can unfold in metaphorically resonant and emotionally attuned ways.

This section details the principles and strategies for designing symbolic environmental architectures in IVR that reflect and facilitate internal psychic processes. The aim is to develop virtual transitional spaces, responsive to affective states, and structured to evoke, mirror, and transform unconscious dynamics.

### 5.1. Symbolic Resonance and the Evocative Function of Space

The virtual environment in analytic VR is not merely a backdrop; it acts as a symbolic landscape that interacts with the user’s emotional world. Drawing from Winnicott’s theory of potential space—a liminal zone between psychic reality and external environment where creativity and subjectivity emerge—the IVR environment becomes a projective screen for unconscious material [56] (see Figure 3).

Designing for symbolic resonance involves constructing spaces that metaphorically correspond to psychic functions: thresholds that mark transitions, corridors that evoke passage and latency, enclosures that represent regression or containment. For instance, a patient with high sympathetic arousal and elevated GSR may encounter a narrowing tunnel with soft textures and dim lighting, inviting both containment and introspective regression [57].

Conversely, expansive landscapes, flooded with light and movement, may facilitate moments of psychic expansion, allowing reorganization of self-representations in cases of depressive withdrawal or dissociation. These environments are not static, but dynamically shaped by real-time biosignals, encoding emotional feedback into the architectural modulation of the space [58] (see Figure 3 for a spatial topography of transitional zones, symbolic thresholds, and archetypal objects within the psychoanalytic VR environment).

**How to read Figure 3:** Zones are ordered from left to right to depict progressive symbolization. (A) Maternal room/regression zone: a protected “holding” area for stabilization; the environment narrows (reduced spatial density, dimmer light) and transitional objects are available. (B) Exploration area: the space gradually opens; textures and soundscape become richer as autonomic arousal stabilizes, inviting curiosity and play. (C) Symbolic threshold: a liminal passage (bridge/arch) signaling readiness for symbolic transformation; crossing occurs only when safety criteria are met (e.g., no high arousal peaks). (D) Symbolic landscape: high-valence symbols (moon, mountains, mandala-like forms) appear and can be approached or left at will; avatar attunement remains non-interpretive (proximity/gaze modulation). Arrows indicate allowed transitions; dashed arrows denote optional returns to the safety area. The layout does not prescribe content; it provides a graded path from containment to exploration to transformation under safety-first rules.

### 5.2. Archetypes, Mythic Imagery, and Rêverie-Driven Elements

To further support symbolic elaboration, environments may incorporate archetypal and mythopoetic imagery—forms that, according to Jungian and post-Bionian traditions, resonate deeply with the collective and personal unconscious [59]. Forests, labyrinths, caves, water bodies, and ancient ruins are not chosen for their realism but for their associative richness, allowing the user to engage symbolically through movement and interaction.

The rêverie function, reimagined here in digital terms, allows the IVR system to modulate these elements in response to the user’s affective profile. For example, when the system detects a decrease in HRV (suggesting emotional dysregulation), a scene may gently morph to introduce symbolic elements of reparation—e.g., a broken bridge mending itself, or fragmented objects reassembling [60].

These dynamic visual metaphors serve as intermediary mental representations, bridging the user’s somatic activation and symbolic elaboration. Importantly, the environment remains ambiguous and underdetermined, avoiding fixed narratives or explicit interpretations, thus preserving the associative freedom essential for unconscious processing [61].

### 5.3. Transitional Objects and Symbolic Interaction

Winnicott’s concept of transitional objects—physical items imbued with subjective meaning and used to negotiate separation and individuation—can be reimagined within IVR through interactable symbolic artifacts. These may include:Floating geometric forms that change shape or color based on user gaze or touchObjects that respond to affective input by growing, disappearing, or transformingNarrative figures (e.g., animals, guides) that remain silent but accompany the user’s path

These objects should be semiotically open, allowing the user to attribute idiosyncratic meaning, project unconscious content, and enact symbolic gestures (e.g., holding, releasing, discarding). Their behavior is modulated by biosensors: elevated GSR may cause the object to glow or emit sound, creating a loop of embodied affective-symbolic interaction [56].

Rather than gamifying the experience, this interactivity serves as symbolic play, a process essential to healing in analytic work. It allows for regression in the service of the ego, enabling contact with pre-verbal layers of experience within a technologically mediated but emotionally safe context.

### 5.4. Environmental Plasticity and the Simulation of Rêverie

The plasticity of the IVR environment—its capacity to shift in real time—supports a digital simulation of rêverie. Rather than representing fixed settings, the system may orchestrate fluid transitions between scenes, morphing textures, symbolic emergence, and subtle aesthetic drift.

This quality parallels the analytic function of rêverie, described by Bion as the capacity to metabolize unprocessed beta-elements into alpha-elements—that is, into material available for thought [62]. In immersive VR, rêverie becomes an aesthetic-affective response, wherein the system itself seems to “dream the user”, adjusting space to meet the emotional state with metaphorical forms.

This environmental rêverie is not merely soothing; it is transformative, allowing the user to symbolically reconfigure internal states. In psychodynamic terms, it constitutes a symbolic act of containment, wherein the VR setting holds, reflects, and modulates unmentalized affect [63].

### 5.5. Soundscapes, Temporal Rhythm, and the Acoustic Holding

Sound design plays a crucial role in symbolic environmental construction. Drawing from psychoanalytic literature on maternal holding and the acoustic container, ambient soundscapes can regulate affect and support symbolic immersion [64].

Low-frequency drones, irregular rhythms, or spatialized echo may serve to externalize internal disorganization, while harmonic progressions and consistent auditory presence may promote affective regulation. In some cases, the soundscape may incorporate bodily-correlated audio events, such as amplified heartbeat or breath sounds, fostering interoceptive awareness and re-embodiment.

Temporal design—the duration and rhythm of events, transitions, and movements—must also be symbolic rather than mechanical. Unexpected delays, slow emergence, or prolonged silence may function as analytic interventions, evoking tension, expectancy, or opening space for meaning.

### 5.6. Symbolic Design as Ethical Practice

Finally, symbolic environmental design must be understood as an ethical commitment. It is not merely about user experience or engagement, but about constructing spaces that respect the user’s emotional vulnerability, symbolic openness, and unconscious depth. Design decisions must avoid suggestive content, overinterpretation, or forced narratives, instead cultivating ambiguity and emotional truth.

By framing the environment as a space of mutual meaning-making, the system affirms the subjectivity of the user while enabling therapeutic transformation.

## 6. Clinical and Technological Implications

The psychoanalytic IVR architecture proposed in this framework offers profound implications for clinical practice, particularly in addressing psychological conditions characterized by symbolic disorganization, affect dysregulation, or resistance to verbal elaboration. Through the integration of IoT-based physiological monitoring and symbolic environmental modulation, this model facilitates a multi-layered therapeutic experience that engages the user not only cognitively and behaviorally but also somatically and symbolically (see Figure 3 for the symbolic processing flow linking biosensor activation, adaptive environmental modulation, avatar-mediated attunement, and emergent clinical insight).

**How to read Figure 4.** Labeled zones represent transitional areas (containment, exploration, threshold crossing). Arrows indicate permissible symbolic transitions under different affective states. Numbered call-outs (A–D) map to: (A) narrowing/holding; (B) symbolic emergence; (C) avatar attunement corridor; (D) safe-room fallback.

However, as with any emerging clinical technology, its implementation must be guided by careful consideration of both its therapeutic potential and ethical, practical, and epistemological limitations.

A high-resolution, numbered version is provided in Appendix A.

### 6.1. Therapeutic Applications and Populations

One of the primary strengths of the proposed IVR system lies in its capacity to simulate and regulate embodied affective states while offering symbolic scaffolding for unconscious elaboration. Such an approach is particularly promising for:(a)Trauma-related disorders

Patients with PTSD, complex trauma, or early attachment disruptions often struggle with implicit emotional memories that are not readily accessible through verbal channels [65]. The capacity of analytic VR to detect physiological signals of distress (e.g., elevated GSR or altered HRV) and modulate the environment accordingly (e.g., reducing sensory load or introducing transitional symbols) offers a non-invasive way to support regulation and integration of traumatic material. Unlike traditional exposure paradigms, this system does not aim for desensitization per se, but rather for symbolic transformation, consistent with psychoanalytic goals [66].

(b)Borderline and narcissistic personality configurations

These clinical structures are marked by difficulties in affect regulation, unstable identity, and primitive defenses. The simulation of stable relational patterns via avatars, combined with emotionally attuned symbolic landscapes, can serve as virtual holding environments, supporting the development of reflective function and internal consistency [67]. Importantly, the avatar’s emotional responses, driven by biometric data, must be regulated to avoid re-traumatization or excessive idealization, both of which are frequent transferential risks in such populations.

(c)Alexithymia and psychosomatic syndromes

Patients who present with limited symbolic access to emotional states or somatization of distress (e.g., fibromyalgia, chronic fatigue, irritable bowel syndrome) may benefit from a system that mirrors somatic patterns and externalizes internal conflict through symbolic spatial dynamics [68]. For example, tightening of the environment in response to physiological tension may facilitate recognition and elaboration of unconscious bodily affect.

### 6.2. Psychoanalytic Integrity and Adaptation

A central concern in translating psychoanalytic concepts into digital environments is the risk of reductionism. The symbolic and unconscious dimensions of psychic life are inherently ambiguous, overdetermined, and resistant to algorithmic codification. Thus, the clinical use of this system must be explicitly positioned not as a substitute for analytic treatment but as a complementary tool, particularly in the preparatory, supportive, or diagnostic phases of psychodynamic therapy [69].

Moreover, the therapeutic function of the analyst cannot be replaced by an avatar or a system. The IVR environment can provide a simulated holding function, but the interpretative, ethical, and empathic dimensions of analytic work must remain with the human clinician. As Lemma argues, the therapist’s presence and the analytic third cannot be virtualized without losing their core function as facilitators of meaning [70].

### 6.3. Ethical and Clinical Safety Considerations

Any implementation of such a system requires robust ethical safeguards. These include:Informed consent regarding physiological data collection, emotional responses, and symbolic content adaptationPsychological screening to identify contraindications (e.g., psychosis, acute suicidality)Clinical supervision and debriefing post-immersion to contextualize symbolic experiencesRespect for symbolic freedom, avoiding predetermined content or forced interpretations

Furthermore, emotional safety protocols must be integrated into the system’s architecture. For example, rapid withdrawal or distress detection can trigger a safe exit from the environment or return to a neutral symbolic zone (e.g., a safe room, transitional space), emulating the therapist’s containing function [71].

### 6.4. Technological Limitations and Future Adaptability

The effectiveness of the proposed system is contingent on several technological constraints:Sensor accuracy and latency: Inaccurate or delayed detection of affective signals could disrupt symbolic coherence or misrepresent user states, leading to confusion or emotional mismatch [72].Hardware comfort and usability: Wearable devices must be non-intrusive and tolerable over time. For patients with tactile sensitivity or somatic concerns, sensor configuration must be adaptable.VR side effects: Immersive systems carry known risks of cybersickness, derealization, and sensorimotor disorientation in sensitive populations. Gradual exposure and session length regulation are essential [73].

Nevertheless, the modularity of the framework allows for incremental implementation, with flexibility to accommodate new biosensors, machine learning for affective pattern detection, and cross-platform integration (e.g., mobile VR, home-based biofeedback units).

### 6.5. Clinical Integration Models

To maximize clinical impact, the IVR system can be integrated into various care settings:In psychotherapy clinics, it may be used before or after analytic sessions, allowing users to explore symbolic content or regulate arousal states.In inpatient units, it can serve as an emotionally attuned relaxation tool, promoting inner organization.In early interventions, it may assist in the symbolization process for children and adolescents with developmental trauma, autism spectrum disorder, or somatic preoccupation.

The system’s symbolic design, grounded in psychoanalytic theory, provides a unique opportunity to bridge somatic regulation and narrative construction, enabling clinicians to explore preverbal dynamics and facilitate embodied insight in patients who may otherwise remain unreachable through verbal interpretation alone [74].

### 6.6. Clinical Pathways and Deployment Scenarios

**Outpatient psychodynamic clinics:** The IVR module is positioned as a 10-min pre-session tool for autonomic regulation and symbolic priming, followed by a 45–50-min psychotherapy session and a 5-min debrief. Safety features include a neutral safety-scene (one-click return) and a panic-exit mapped to the handheld controller. Feasibility and acceptability are monitored with presence (IPQ), cybersickness (SSQ), and a therapist-rated alliance indicator; biosignal logs are de-identified and summarized at the session level (e.g., HRV trend, EDA peaks), without storing raw audio/video.

**Inpatient trauma units:** Sessions are shorter (8–10 min), delivered twice weekly under nursing supervision, with strict stop-rules and real-time monitoring of autonomic arousal. The symbolic environment defaults to low-stimulus “holding” scenes with gradual transitions. Outcomes focus on tolerability, arousal trends (e.g., reduction in phasic EDA peaks across sessions), and a pragmatic sleep proxy (patient-reported sleep continuity). All optional modules (e.g., prosody) are disabled by default; re-enablement requires explicit opt-in and clinical justification.

**Adolescents:** Scenes are briefer (6–8 min), emphasize low-directive transitional objects, and are preceded by a caregiver briefing and explicit consent. The pathway prioritizes engagement and affect labeling over insight, with graduated challenges (from containment to exploration) and the ability to revert instantly to a safety-scene. Process indicators include presence/tolerability, simple engagement metrics (time on task, approach–avoidance balance), and a brief therapist-rated alliance check.

### 6.7. Clinical Validation Criteria

Beyond technical feasibility (latency, packet-loss) and tolerability (SSQ), we specify three clinical criteria to evaluate the framework in proof-of-concept and pilot phases:Safety and Tolerability: No adverse events; session-completion ≥90%; Simulator Sickness Questionnaire (SSQ) below moderate range; neutral safety-scene effectively aborts sessions on demand.Working Alliance and Acceptability: A brief therapist-rated alliance indicator and a 1–2 item patient acceptability check collected after each session (e.g., perceived helpfulness, willingness to repeat). Target: stable or improving scores across sessions.Affective Regulation and Presence: Physiological trends consistent with containment (e.g., ↑RMSSD, ↓phasic EDA peaks during holding scenes), alongside presence indices (IPQ/ITC-SOPI) in the acceptable range without cybersickness escalation.

Exploratory process markers (optional) include approach–avoidance balance, time on task, and frequency of returns to the safety scene. Responders can be provisionally defined by concurrent improvement in at least two domains (tolerability, alliance/acceptability, autonomic regulation).

## 7. Discussion

The integration of psychoanalytic constructs into immersive virtual reality (IVR) platforms, as outlined in this conceptual model, invites a rethinking of how symbolic, unconscious, and relational dimensions can be meaningfully incorporated into technologically mediated therapeutic environments. At its core, this approach challenges the dominant epistemology of current digital mental health technologies, which are often grounded in cognitive-behavioral paradigms emphasizing explicit behaviors, rational decision-making, and symptom reduction [75].

While cognitive behavioral therapy (CBT) and exposure-based VR protocols have demonstrated efficacy for anxiety, phobias, and PTSD, they typically function within a stimulus-response framework, aiming to extinguish maladaptive associations through repeated exposure [76]. In contrast, the psychoanalytic IVR model outlined here does not rely on habituation or desensitization but seeks to activate, mirror, and transform unconscious processes through symbolic modulation, affective attunement, and relational simulation. This marks a fundamental shift from a behaviorist logic to a metapsychological and intersubjective one, in which the user’s embodied subjectivity and affective depth are central.

Furthermore, the model is epistemologically aligned with enactive and embodied theories of cognition, which assert that perception, action, and emotion emerge from recursive interactions between the body and the environment [77]. Rather than viewing physiological data as mere markers of arousal, the model treats biosignals as carriers of unconscious meaning, amenable to symbolic elaboration within a digitally constructed therapeutic holding space.

### 7.1. Comparison with Existing Technological Frameworks

Current affective computing systems often strive to decode emotion from biometric inputs, translating physiological changes into pre-labeled emotional states such as “fear,” “joy,” or “anger.” These taxonomies, while useful in commercial or behavioral applications, fail to capture the polysemic, ambivalent, and intersubjective nature of human affect as understood in psychoanalysis [78]. The psychoanalytic model, by contrast, does not aim to interpret or categorize emotions algorithmically but to facilitate the user’s own symbolic transformation of affect within a space of ambiguity and resonance.

Other emerging frameworks in VR-based psychotherapy have begun to explore embodied cognition and self-representation (e.g., using avatars that reflect the user’s postural or facial expressions), yet they frequently lack a symbolic or psychodynamic dimension [79]. Our model adds this missing layer by positioning symbolic resonance, transitional phenomena, and transferential dynamics as structural components of the virtual environment, thereby aligning the system’s logic with the foundational principles of analytic treatment.

Additionally, while some VR interventions incorporate narrative-based scenarios, these are usually scripted, linear, and goal-directed. In contrast, the proposed system promotes open-ended, emergent narratives, shaped dynamically by unconscious cues and user-initiated meaning-making. This reflects the psychoanalytic emphasis on free association, the avoidance of directive interventions, and the privileging of subjective interpretation over externally imposed structure [80].

### 7.2. Clinical Implications of an Epistemological Shift

The epistemological repositioning of the VR environment as a symbolic container, rather than a behavioral training tool, has significant clinical implications. First, it recognizes that therapeutic change is not solely a matter of behavior modification but often involves the integration of dissociated affects, the transformation of primitive defenses, and the symbolization of unrepresented psychic states [81].

Second, the model offers an alternative to neurobiological reductionism in digital psychiatry, which tends to conceptualize affect as a neurochemical event to be corrected pharmacologically or algorithmically. Instead, it restores a meaning-centered approach, where biosignals are not treated as pathologies to be normalized but as expressions of subjective experience to be understood, held, and transformed [82].

This model also supports therapeutic pluralism, offering a platform that can complement analytic psychotherapy, especially in populations where verbal access is limited or symbolic function is underdeveloped (e.g., in early trauma, somatic conditions, or alexithymia). By engaging the body, the senses, and symbolic imagination simultaneously, the system enables non-verbal access to internal states, often unreachable through language alone [83].

### 7.3. Future Perspectives and Theoretical Integration

The proposed architecture invites further exploration at the intersection of psychoanalysis, bioengineering, and immersive technology. Its development may benefit from dialogue with related disciplines, such as phenomenological psychopathology, neuropsychoanalysis, and somatic therapies, which share a commitment to integrating bodily processes and subjective meaning [84].

Moreover, this model resonates with recent calls in digital mental health to move beyond mere efficiency and scalability toward ethically grounded, relational, and culturally sensitive design [85]. It argues for a form of digital therapy that does not circumvent complexity, but rather embraces the opacity, uncertainty, and creativity that define human psychological life.

## 8. Conclusions and Future Directions

This article has proposed a conceptual framework for the development of psychoanalytically informed immersive virtual reality (IVR) environments, integrated with IoT-based biosensing technologies. Departing from the dominant paradigms of cognitive-behavioral VR interventions, the model outlined herein emphasizes symbolic modulation, affective mirroring, and intersubjective simulation, drawing upon foundational psychoanalytic constructs such as rêverie, transference, and the analytic setting. The architecture is structured in three recursive layers—somatic–sensorial input, symbolic–narrative modulation, and relational interface—which collectively aim to create a dynamic digital container for unconscious processes.

The clinical implications of such a model are far-reaching. It holds promise for populations with early relational trauma, alexithymia, or psychosomatic expression, where traditional verbal psychotherapy may be insufficient to access and transform implicit emotional content. Moreover, it offers a new modality of therapeutic presence that, while not replacing the analyst, can simulate holding, resonance, and containment in a technologically mediated space. The system’s real-time responsiveness to biosignals allows it to operate as a transitional object, enabling symbolic experimentation and embodied insight.

From a technological standpoint, this model invites further interdisciplinary collaboration between clinicians, designers, and engineers, particularly in refining affective computing algorithms to remain ethically grounded and symbolically open-ended. Avoiding the reduction of complex affective states into narrow interpretative categories remains an epistemological priority, in line with the psychoanalytic commitment to uncertainty and ambiguity as preconditions for psychic elaboration.

Future research should explore the empirical validation of such systems through mixed-method trials, combining psychophysiological measures with qualitative analyses of user experience. Special attention should be paid to identifying indications and contraindications, defining the therapist’s role in co-facilitating VR experiences, and ensuring integration with broader treatment plans. Additionally, open questions remain about the system’s applicability across developmental stages, cultural contexts, and diagnostic spectra.

This work advances a foundational conceptual architecture rather than reporting experimental results. To bridge concept and practice, we provide a minimal implementation sketch and a staged validation plan (bench → POC → pilot) with explicit safety and privacy safeguards. The framework is intended as an open, ethically grounded starting point for interdisciplinary collaborations between engineers and psychodynamic clinicians.

## Figures and Tables

**Figure 1 bioengineering-12-01257-f001:**
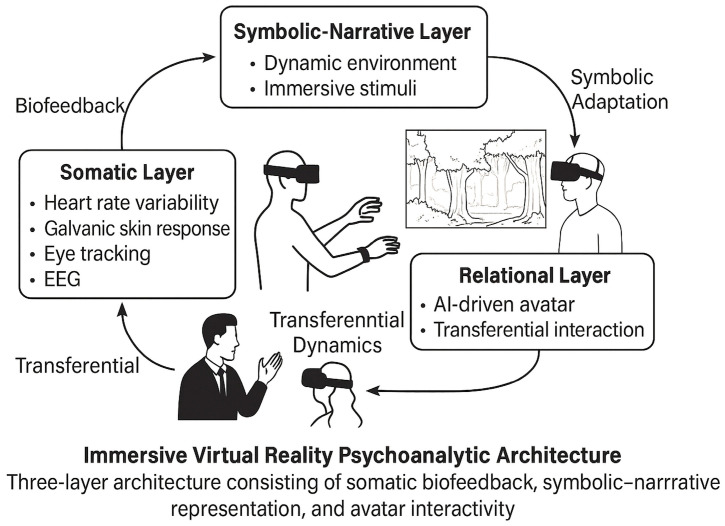
Three-Layer Conceptual Model Integrating Immersive VR, Biosensor Interfaces, and Psychoanalytic Functions. Legend: The architecture comprises somatic–sensorial inputs (HRV, EDA/GSR, EEG, eye-tracking, respiration), symbolic–narrative modulation (light, space, soundscape, transitional objects), and relational interface (avatar attunement). Arrows indicate recursive feedback; dashed outlines denote optional modules.

**Figure 2 bioengineering-12-01257-f002:**
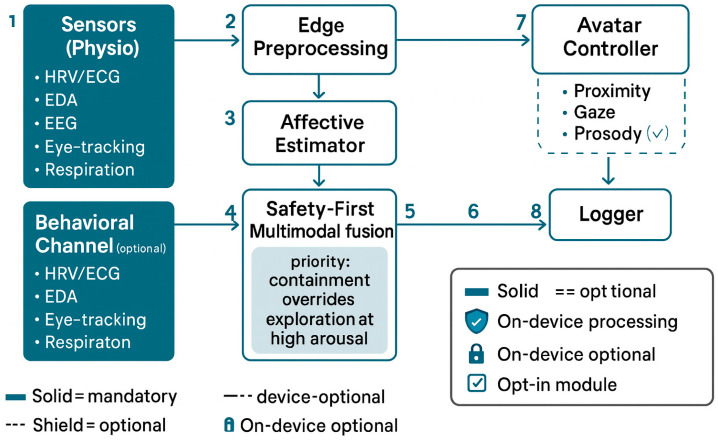
System Architecture with Behavioral Channel and I/O Flows. Legend: Physiological and behavioral streams are preprocessed on-device and fused under safety-first rules to drive symbolic environment parameters and avatar attunement. Optional modules (e.g., prosody) are dashed and disabled by default; logging stores de-identified session summaries only. 1—Sensors (Physio)—acquisition of primary biosignals; 2—Edge preprocessing—on-device cleaning/segmentation of raw streams; 3—Affective estimator—inference of latent states from the preprocessed features; 4—Behavioral channel (optional)—optional behavioral inputs routed directly into the fusion step to complement physiologic data; 5—Output of Safety-First multimodal fusion → Mapping Engine—marker on the right edge of the fusion block indicating the hand-off of fused affective signals to the symbolic/environment mapping engine; 6—Pipeline segment → VR runtime—marker placed on the horizontal line to indicate transmission of mapped parameters to the VR runtime for scene/parameter updates; 7—Avatar controller—module that drives avatar behaviors; 8—Logger—de-identified session logging/audit trail receiving events from the control loop.

**Figure 3 bioengineering-12-01257-f003:**
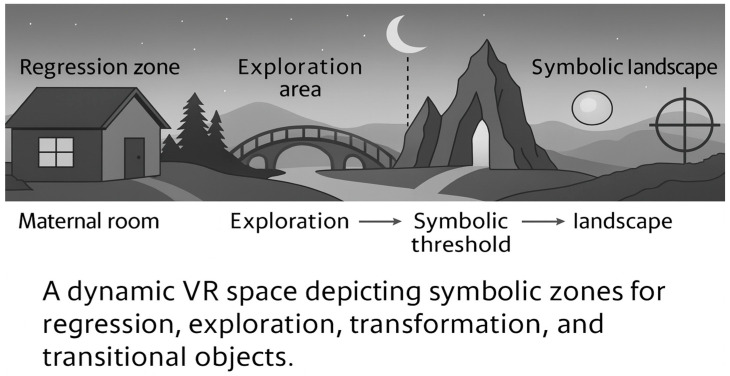
Transitional Space Topography. Legend: This conceptual map illustrates the spatial organization of a psychoanalytically-informed virtual environment. Transitional zones, symbolic thresholds, and archetypal objects are dynamically modulated to mirror the patient’s internal world.

**Figure 4 bioengineering-12-01257-f004:**
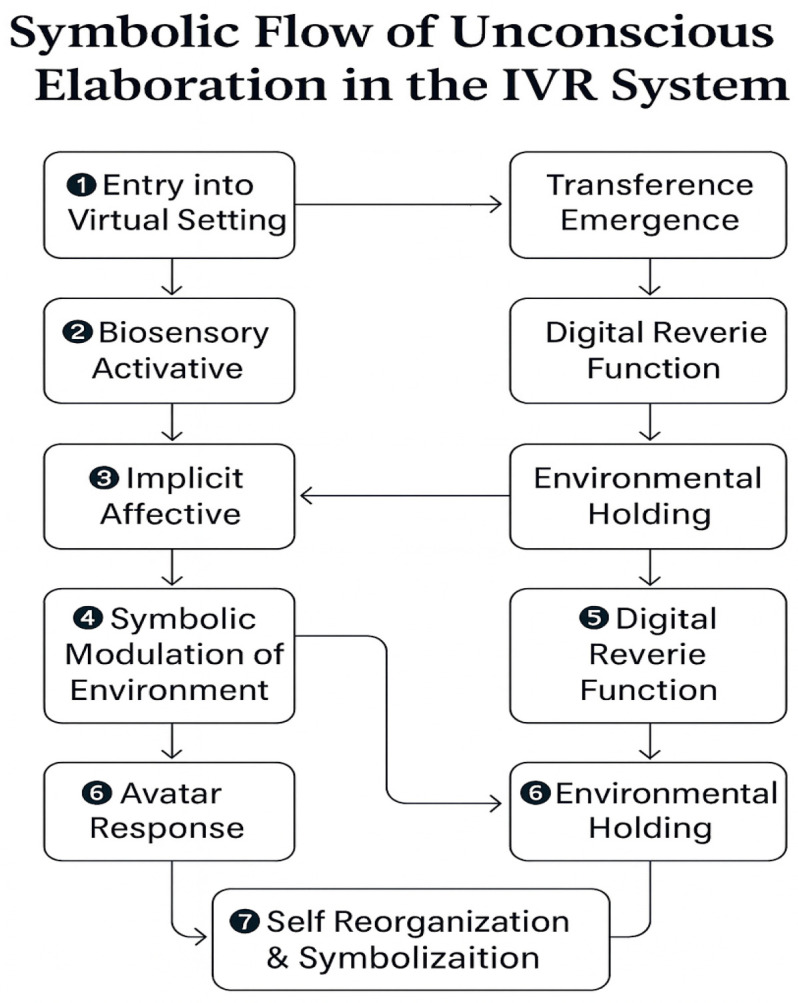
Symbolic flow of unconscious elaboration in the immersive virtual reality system. Legend: A seven-step closed loop: (1) multimodal acquisition; (2) edge preprocessing; (3) affect proxies (vagal withdrawal, sympathetic arousal, avoidance); (4) safety-first fusion; (5) symbolic environment mapping; (6) avatar adaptation; (7) logging/recursive update. Solid = mandatory; dashed = optional.

**Table 1 bioengineering-12-01257-t001:** Mapping Between IoT Biosignals/Behavioral Features and Psychoanalytic Functions with Quantified Modulation Parameters.

Signal/Feature	Physio/Behavioral Marker	Psychoanalytic Interpretation	Environmental/Avatar Modulation (Quantified)
HRV (↓ RMSSD)	Low vagal tone; sympathetic dominance	Regression, anxiety, impaired self-soothing	LightIntensity −20%; SpaceDensity +15%; AvatarProximity +0.3 m; AvatarMotion −25%
EDA (↑ phasic peaks/tonic SCL)	Sympathetic arousal	Unprocessed affect; somatic projection	TransitionalObject = ON; Sound low--pass; ColorTemp 3500 K; Prosody rate −15%
EEG alpha asymmetry (R > L)	Withdrawal/dysphoria bias	Relational withdrawal; depressive tone	Soundscape Soothing; ColorTemp 4000 K; AvatarGaze averted; SceneContrast −10%
Eye-tracking (avoidance/over-fixation; blink rate)	Gaze aversion or hyper-fixation	Resistance; hypercathexis	SymbolOpacity ±20%; PathVisibility toggle; AvatarProximity +0.2 m when avoidance > θ
Respiration (shallow/irregular)	Hyperventilation; breath--holding	Defensive constriction; repression	BreathPacing 6 bpm cue; Light pulsation ±10%; SpaceDensity −10%
Head/hand kinematics (variability ↓; micro-pauses ↑)	Motoric freezing/avoidance	Inhibition; fear; retreat	SpaceDensity −15%; LightIntensity −10%; AvatarProximity +0.4 m
Approach–avoidance distance (avatar/objects)	Relational withdrawal or engagement	Attachment dynamics; testing of safety	PathWidth ±30%; AvatarGaze on/off; AvatarProximity ±0.3 m
Interaction tempo (object manipulation)	Hyper/hypo engagement	Agitation vs. apathy	SymbolicObject spawn/despawn (±50%); Soundscape ±6 dB; TaskHints on/off

Legend: The fourth column reports concrete IVR parameters (relative % or absolute units) applied under safety-first rules; containment overrides exploration when arousal is high. Notes: Symbolic modulation strategies are knowledge-driven (setting, rêverie, transitional phenomena) and grounded in empirical biosignal literature—HRV metrics and norms [38] electrodermal activity [39]; EEG frontal alpha asymmetry [40]; gaze-based mind-wandering/avoidance [41]; respiration–emotion coupling [42]. Behavioral features are fused with physiology under safety-first rules. ↑/↓ = increase/decrease relative to the individualized baseline for that signal (operationally: deviation from the subject’s rolling baseline; e.g., a meaningful change ≈ |z| ≥ 1). ± = bidirectional adjustment around the current/default scene value (e.g., “±20%” means relative up/down modulation of the parameter).

## Data Availability

No new data were created or analyzed in this study. Data sharing is not applicable to this article.

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
