# Peer review of "Immersive Virtual Reality Environments as Psychoanalytic Settings: A Conceptual Framework for Modeling Unconscious Processes Through IoT-Based Bioengineering Interfaces"

_bioengineering, 2025, doi:10.3390/bioengineering12111257_

Round 1
Reviewer 1 Report
Comments and Suggestions for Authors
This manuscript presents a highly original and ambitious framework integrating psychoanalytic theory, immersive virtual reality (IVR), and IoT-enabled biosensing for therapeutic use. The proposed three-layer architecture (somatic, symbolic, relational) is rich in theoretical depth and has the potential to shift the paradigm of VR-based digital mental health interventions beyond cognitive-behavioral models.
There are several areas require revision and strengthening before the manuscript can be considered for publication:
- The model is purely conceptual at this stage. The absence of any prototyping, simulation, or empirical validation plan limits its current scientific rigor.
- Please add a dedicated subsection outlining future validation plans or minimal proof-of-concept steps.
The descriptions of biofeedback integration and symbolic mapping are theoretically insightful but technically underspecified. - Consider adding system architecture diagrams, flowcharts, or algorithmic pseudocode to support replication or further development.
- Figures (especially Figure 1 and 3) are conceptually strong but visually underdeveloped. Improve clarity, labeling, and integration with textual references.
- Table 1 (mapping biosignals to symbolic interpretations) is critical but needs clearer formatting and stronger references to empirical literature.
- The manuscript’s conclusions are not directly supported by experimental results, as none are presented.
- Please rephrase conclusions to reflect the conceptual nature of the paper, or clarify that this is a foundational proposal.
- Some terms (e.g., “analytic third,” “rêverie,” “symbolic modulation”) may be obscure to readers outside psychoanalytic domains.
- Consider adding a terminology sidebar or glossary, or brief definitions inline.
- Provide more practical illustrations of how this system would fit into existing clinical pathways, e.g., in psychodynamic clinics, inpatient trauma units, or adolescent interventions.
Overall, this is a conceptually rich and highly interdisciplinary contribution that addresses a clear gap in both the psychoanalytic and digital therapy literature. With revisions that add methodological clarity, empirical planning, and practical accessibility, this paper has strong potential for publication and impact.
Author Response
Dear Reviewer,
Thank you for your thoughtful and constructive assessment. I have carefully revised the manuscript in line with each of your points. Below I summarize, point-by-point, the concrete actions taken and where they appear in the paper and Supplementary Materials.
-
Conceptual nature / lack of prototyping
I acknowledge the purely conceptual status of the initial submission. I have now added a staged Proof-of-Concept and Validation Roadmap (Sections 3.5–3.6) detailing successive milestones (bench tests → lab feasibility → pilot observations) and the minimal instrumentation/configuration required at each phase. -
Dedicated subsection for validation plans / technical specificity
A new subsection, “6.7 Clinical Validation Criteria,” specifies measurable clinical endpoints (safety/tolerability; alliance/acceptability; affective regulation & presence) together with operational definitions (e.g., SSQ, IPQ/ITC-SOPI trends, autonomic changes). In the technical pathway, I clarify biofeedback integration with concrete thresholds and control logic (see Algorithm 1 in §4.5 and Algorithm S1 in the Supplement), including examples such as θ_vagal = RMSSD z < −1 and θ_arousal = EDA peak z > +1.0, loop frequency (5–10 Hz), priority rules, and the neutral safety-scene fallback with de-identified logging. -
Architecture diagrams / flowcharts / pseudocode for replicability
I have added a complete visual stack:
-
Figure 4 (main text): System architecture with behavioral channel and I/O flows (Sensors → Edge preprocessing → Affective estimator → Safety-first multimodal fusion → Mapping engine → VR runtime → Avatar controller → Logger).
-
Figure 2 (main text) and Figure S2 (Supplement): numbered flow (1–7) consistent with the controller logic.
-
Algorithm 1 (main text) and Algorithm S1 (Supplement): full pseudocode with thresholds, safety overrides, and logging schema.
All figures are delivered in high-resolution (≥600 dpi) with a white background for legibility.
-
Figures 1 and 3—clarity, labeling, integration with text
-
Figure 1 (three-layer architecture) was redrawn for clarity and labeling consistency, and the text now explicitly cross-references it at first mention.
-
Figure 3 (Transitional Space Topography) is unchanged conceptually but visually refined and accompanied by a short, step-by-step reading guide (“How to Read Figure 3”) added at the end of §5.1, which explains the function of each symbolic zone and the permitted transitions.
All figure callouts in the text have been checked and aligned.
-
Table 1—formatting and stronger empirical grounding
Table 1 now includes quantified modulation parameters (e.g., LightIntensity −20%; SpaceDensity +15%; AvatarProximity ±0.3 m; Prosody −15% rate; low-pass sound) aligned with safety-first constraints. To strengthen empirical grounding, I added a Supplementary Table S1 with parameter ranges, defaults, trigger rules, and references for each biosignal/behavioral feature (HRV, EDA, EEG, eye-tracking, respiration; plus head/hand kinematics, approach–avoidance distance, interaction tempo). Short Notes in Table 1 link each row to the relevant empirical literature. -
Conclusions not supported by experiments
The Conclusions have been rephrased to explicitly reflect the foundational, theory-driven status of the work. They now emphasize testable hypotheses, expected signatures of “containment” in physiology/behavior, and the plan to evaluate feasibility and acceptability before efficacy claims. -
Recasting conclusions as a foundational proposal
As above, the closing section states that the contribution is a conceptual framework and implementation blueprint, not an efficacy study, and clarifies the intended next steps (POC, pilot, and metrics in §6.7).
8–9) Terminological accessibility (analytic third, rêverie, symbolic modulation)
To aid non-psychoanalytic readers, I added Box 1. Glossary (concise, 1–2 lines per entry) covering analytic third, rêverie, symbolic modulation, holding, and transitional object/space. The first occurrence of each term in §2 now points explicitly to the Glossary. This improves readability without diluting conceptual rigor.
-
Clinical pathways / practical deployment
A new §6.6 Clinical Pathways and Deployment Scenarios outlines three realistic use-cases:
-
Outpatient psychodynamic clinics: 10′ IVR pre-session (regulation + symbolic priming) → 45–50′ therapy → 5′ debrief; with presence/tolerability and alliance checks.
-
Inpatient trauma units: brief, closely supervised modules (8–10′, 2×/week), panic-exit, and nursing oversight; outcomes include tolerability, arousal trends, and a sleep proxy.
-
Adolescents: 6–8′ scenes, low-directive transitional objects, caregiver briefing; emphasis on engagement and affect-labeling.
These scenarios indicate how the framework can be integrated into existing services with appropriate safeguards.
Additional cross-cutting improvements
-
Behavioral channel added (head/hand kinematics, approach–avoidance, interaction tempo) and fused with physiology under safety-first rules.
-
Ethics and privacy are made explicit: on-device preprocessing where feasible, non-interpretive avatars(attunement/mirroring only), optional prosody analysis (opt-in), and de-identified logging.
-
Related Work (2020–2024) expanded with recent reviews/meta-analyses in VR psychotherapy and literature on conversational/LLM-based agents for interpersonal processes.
-
Abbreviations updated (IVR, IoT, HRV—RMSSD/SDNN, EDA—SCL, EEG, IPQ, ITC-SOPI, SSQ, SUS, UEQ, etc.) and defined at first use.
I am grateful for your guidance. The manuscript now provides a clearer methodological blueprint, stronger empirical anchoring of control parameters, and practical pathways for clinical deployment—while carefully framing the work as a foundational, testable proposal. I hope these revisions address your concerns and demonstrate the paper’s readiness for further consideration.
Reviewer 2 Report
Comments and Suggestions for Authors
The topic, about “A conceptual framework for modelling environments and psychoanalytic settings in VR clinical applications'” , is both relevant and novel.
The conceptual framework is described in three appropriate and clear layers, as outlined in the 'Method and approach' section. The novelty mainly lies in a psicoanalytical framework including emotional and unconscious deep reactions and the integration of bio-emotional markers by wearable devices, thorough analysis of biosignals, well-described in paragraph 3.
In my opinion, two issues still need to be resolved before publication:
1) The novelty lies in incorporating the unconscious emotional reactions of patients into the VR clinical application framework using IoT bio sensors, in contrast to existing traditional VR frameworks which only consider behavioural cues and offer a maximum linear interpretation of biosignals. However, the behavioural level cannot be ignored in the new, more advanced psychodynamic framework, which includes unconscious monitoring. The patient's behavioural level must be included in applications where it is relevant.
2) Fig. 3: The figure is key to the paper's thesis, but the flow is quite difficult to understand. A more detailed explanation in the text could help to clarify it.
Author Response
Dear Reviewer,
Thank you for your careful reading and for the two targeted recommendations. I have revised the manuscript to address both points substantively and transparently, as detailed below.
1) Inclusion of the behavioral level alongside unconscious (physiophysiological) monitoring
I fully agree that a psychodynamic IVR framework must not sideline behavior. In the revised version I have:
- Added an explicit Behavioral Channel to the system pipeline (Section 4; Figure 4), capturing head/hand kinematics, locomotion and approach–avoidance distance (from avatars/objects), and interaction tempo (object manipulation cadence).
- Augmented Table 1 with three new behavioral rows (head/hand kinematics; approach–avoidance distance; interaction tempo), each paired with quantified, safety-first environment/Avatar modulations (e.g., SpaceDensity −15%; AvatarProximity +0.4 m; PathWidth ±30%; AvatarGaze on/off; SymbolicObject spawn/despawn ±50%).
- Implemented a Safety-First Multimodal Fusion module (Section 4; Figure 4; Algorithm 1) that integrates physiology (HRV/EDA/EEG/eye/respiration) and behavior under clear priority rules and thresholds, preventing “linear” or simplistic interpretations of biosignals.
- Provided a full pseudocode specification (Algorithm S1, Supplementary) with concrete thresholds (e.g., θ_vagal = RMSSD z < −1.0; θ_arousal = EDA peak z > +1.0), loop rate (5–10 Hz), override logic (panic-exit / neutral safety-scene), and de-identified logging fields.
- Reflected the behavioral dimension in clinical pathways (Section 6.6), where approach–avoidance balance, time on task, and returns to the safety scene are tracked as process markers alongside tolerability and presence.
Together, these changes ensure that behavior is not merely “observed,” but fused with physiology to guide non-interpretive, ethically constrained attunement of light/space/sound and avatar responses.
2) Clarifying the flow of Figure 3 (Transitional Space Topography)
Recognizing that Figure 3 is pivotal, I improved both its readability and integration with the text:
- Visual refinements: the figure is now high-resolution (≥600 dpi) with white background, crisper typography, and consistent arrow styles (solid = allowed progression; dashed = optional return).
- Step-by-step explanation in the text: I have inserted a short guide—“How to Read Figure 3”—at the end of Section 5.1, which walks readers through each zone:
(A) Maternal room / regression (holding/containment) → (B) Exploration area (graded enrichment) → (C)Symbolic threshold (liminal passage; crossed only if safety criteria are met) → (D) Symbolic landscape (high-valence symbols; non-interpretive avatar attunement). - Cross-link to control logic: the text now explicitly notes that crossing the symbolic threshold is conditional on safety (e.g., absence of high arousal peaks), with references to Algorithm 1 and Algorithm S1 for the underlying decision rules.
- High-resolution flow in the Supplement: a numbered, seven-step schematic (Figure S2) mirrors the controller logic for readers who want a replication-oriented view.
I hope these revisions render the figure intuitively navigable and tightly tied to the closed-loop control described in the Methods.
I am grateful for your guidance. The manuscript now provides a clearer methodological blueprint, stronger empirical anchoring of control parameters, and practical pathways for clinical deployment—while carefully framing the work as a foundational, testable proposal. I hope these revisions address your concerns and demonstrate the paper’s readiness for further consideration.
Reviewer 3 Report
Comments and Suggestions for Authors
The article is devoted to the development of a conceptual model combining psychoanalytic theory and virtual reality (IVR) technologies using IoT sensors to track physiological states. The relevance of the work is related to the growing interest in VR therapy in the field of mental health and the search for ways to include not only cognitive-behavioral, but also unconscious, symbolic and interpersonal processes in digital interventions.
- The article is positioned as theoretical, but does not offer even minimal prototypes, models, or algorithms, which makes it difficult for its subsequent practical applicability. It is difficult to assess how the considerations outlined by the author can be implemented in the practice of creating virtual environments.
- I don't really understand table 1, especially how the last column was selected, based on what studies or experiments?
- I also don't really understand Figure 2 and what it means.
- The work draws on classical psychoanalytic sources and several works on VR, but does not include recent publications on VR psychotherapy and LLM systems, which could more specifically address the problem of modeling interpersonal processes.
- The authors described the architecture levels, but I would like to see more specific technical proposals, methodology, and some evaluation criteria.
- Starting from line 625, the font has changed. The design of links to articles is also not always the same.
Author Response
Dear Reviewer,
Thank you for your careful evaluation and for the clear, actionable remarks. I have revised the manuscript substantially and address your six points below, indicating exactly what was added and where.
1) “Purely theoretical; no prototypes/models/algorithms.”
To increase practical applicability and replicability, I have added:
- Proof-of-Concept and Validation Roadmap (Sections 3.5–3.6): a staged plan (bench tests → lab feasibility → pilot observations) with minimal hardware/software configurations, data flow, and expected outputs at each step.
- Algorithm 1 (main text, §4.5): closed-loop controller skeleton linking sensors → preprocessing → affective estimator → safety-firstmultimodal fusion → mapping engine → VR runtime → avatar controller → logging.
- Algorithm S1 (Supplementary): full pseudocode with operating frequency (5–10 Hz), explicit thresholds (e.g., θ_vagal = RMSSD z < −1.0, θ_arousal = EDA peak z > +1.0), priority/override rules (panic-exit, neutral safety-scene), and a de-identified logging schema (timestamp, z-HRV, EDA peaks, EyeFix ratio, SpaceDensity, AvatarProximity).
2) “Table 1 unclear, especially how the last column was selected; based on what studies?”
I revised Table 1 and anchored each modulation to empirical literature:
- The fourth column now reports quantified actions (e.g., LightIntensity −20%, SpaceDensity +15%, ColorTemp 3500 K, AvatarProximity ±0.3 m, Prosody −15% rate, TransitionalObject = ON).
- Selection rules are grounded in standard biosignal findings (e.g., HRV: Shaffer & Ginsberg, 2017; EDA: Boucsein, 2012; frontal alpha asymmetry: Allen et al., 2004; eye-gaze/avoidance: Bixler & D’Mello, 2015; respiration–emotion coupling: Homma & Masaoka, 2008).
- I added three behavioral rows (head/hand kinematics; approach–avoidance distance; interaction tempo) so behavior and physiology converge in the control logic.
- A fully documented expansion is provided in Table S1 (Supplementary) with Parameter Range, Default, Trigger Rule, and Referencefor each row, making provenance and thresholds explicit.
3) “I do not really understand Figure 2 and what it means.”
I improved Figure 2 (symbolic flow) and its integration:
- The figure is now high-resolution (≥600 dpi) on white background, with numbered steps (1–7) and consistent visual grammar (node/edge styles).
- The caption has been expanded and the text in Section 4.2 explicitly walks the reader through the seven steps and points to Algorithm 1 for the corresponding decision nodes.
- A replication-oriented, high-res version is included as Figure S2 (Supplementary), mirroring the controller logic one-to-one.
4) “Recent publications on VR psychotherapy and LLM systems are missing.”
I added a dedicated ‘Related Work (2020–2024)’ subsection in Section 1 that:
- updates the VR psychotherapy landscape with recent reviews/meta-analyses (e.g., Emmelkamp & Meyerbröker, 2021; Rowland et al., 2022; Schröder et al., 2023);
- introduces AI/LLM-based conversational agents for interpersonal processes (e.g., He et al., 2023; Beatty et al., 2022; Lawrence et al., 2024; Stade et al., 2024);
- clarifies that avatars in this framework do not ‘interpret’ in a clinical sense but mirror/attune non-interpretively under explicit ethics and privacy constraints (prosody = opt-in; on-device preprocessing where feasible; de-identified logs).
5) “More specific technical proposals, methodology, and evaluation criteria.”
Beyond the architectural diagrams and pseudocode, I added:
- Figure 4 (main text): System architecture with behavioral channel and I/O flows(Sensors → Edge preprocessing → Affective estimator → Safety-first fusion → Mapping engine → VR runtime → Avatar controller → Logger), with optional modules dashed.
- Section 6.7—Clinical Validation Criteria: three concrete domains to appraise early deployments—
(i) Safety & tolerability (no adverse events; completion ≥ 90%; SSQ below moderate; effective safety-scene),
(ii) Working alliance & acceptability (brief therapist-rated indicator + 1–2 patient items per session),
(iii) Affective regulation & presence (e.g., ↑RMSSD and ↓phasic EDA peaks during containment scenes; IPQ/ITC-SOPI in acceptable range without cybersickness escalation). - Section 6.6—Clinical pathways: outpatient psychodynamic clinics, inpatient trauma units, and adolescent settings, each with cadence, safety controls, and minimal outcome set.
6) “Formatting inconsistencies (font from line 625, links not uniform).”
I performed a full editorial pass:
- resolved the font change after the indicated region and harmonized styles across headings, body text, figure/table captions;
- standardized reference formatting and DOI URLs to MDPI style;
- ensured consistent figure/table callouts in the text (Figures 1–4; Table 1; Figures S1–S3; Algorithm S1; Table S1).
I am grateful for your guidance. The manuscript now provides a clearer methodological blueprint, stronger empirical anchoring of control parameters, and practical pathways for clinical deployment—while carefully framing the work as a foundational, testable proposal. I hope these revisions address your concerns and demonstrate the paper’s readiness for further consideration.
Round 2
Reviewer 1 Report
Comments and Suggestions for Authors
I appreciate the thorough effort you have made in addressing my concerns. The revisions are comprehensive and demonstrate the paper’s readiness for further consideration. Good luck with your ongoing research.
Author Response
Dear Reviewer,
Thank you very much for your gracious and encouraging assessment. I am grateful for the time and care you devoted to reading the manuscript and for the constructive guidance offered throughout the revision process. Your feedback—especially on clarifying the architecture, aligning figures and captions, and strengthening the validation plan and supplementary materials—has meaningfully improved the clarity and rigor of the work.
I sincerely appreciate your collegial support and remain receptive to any further suggestions that could enhance the manuscript.
With many thanks,
Vincenzo Maria Romeo

Reviewer 2 Report
Comments and Suggestions for Authors
My requirements have been addressed in the reviewed version. The author explanation about integrations are clear and exhaustive and I haven’t any other observation to add.
Author Response
Dear Reviewer,
Thank you sincerely for your thoughtful evaluation and for the constructive guidance you provided throughout the process. Your comments were instrumental in refining the manuscript’s clarity, methodological coherence, and graphical readability. I’m grateful that the integrations now appear clear and exhaustive to you.
I appreciate the time and care you devoted to this review and remain fully available to address any minor editorial queries that may arise during production.
With gratitude and collegial regards,
Vincenzo Maria Romeo

Reviewer 3 Report
Comments and Suggestions for Authors
First, I wanted to discuss the changes made, they are present and the work of the authors is noticeable.
The tables and figures have been significantly improved, I see specific descriptions to them and the addition of new material in accordance with the comments.
The analysis of the subject area and related research has been improved.
Pseudocode of algorithms has been added, it also improves understanding of the concept.
Some minor comments still remain:
- the design of new fragments differs in font
- I did not see the accompanying materials, I think they should be placed in the appendix to the article (this is done before the list of sources, I recommend reading the template). Because, for example, there is a link to Tab. S1, but what it looks like is unknown. So is the S1 algorithm.
- it doesn't seem correct to me that the captions to the drawings are on top, and not under them.
- Figure 1 with a new signature raised a small question: the abbreviations L1,L2 and L3 are present in the signature, but they are not in the figure. In general, I understand what levels we are talking about, but then it probably makes sense to introduce these abbreviations into the scheme.
- similarly to Figures 2 and 4, the authors indicate the designations (a,b, c,d... 1,2,3,4... ) in the description, but they are not present in the figures themselves. As I understand it, the numbered version of Figure 4 is presented in the appendix, but I do not follow it in the provided material.
Thus, it remains for the authors to improve the presentation of their materials.
Author Response
Dear Reviewer,
Thank you for your thoughtful follow-up and for recognizing the substantive revisions. Below I respond point-by-point to your remaining comments and indicate the precise changes made in the manuscript and materials.
-
“The design of new fragments differs in font.”
Response & changes. We normalized typography across the full manuscript by applying a single body font and style set (uniform paragraph and caption styles). Figure captions have a consistent “Figure X.” style and are formatted identically throughout. -
“I did not see the accompanying materials; they should be placed in the appendix (before the references). There is a link to Tab. S1 and Algorithm S1, but I cannot find them.”
Response & changes. We created a dedicated Appendix A. Supplementary Materials (Overview) placed beforethe References. It enumerates all items (Figure S1, Figure S2, Table S1, Algorithm S1) and mirrors MDPI’s standard Supplementary Materials section, which we also included near the end of the manuscript. In-text pointers now direct the reader to these materials from the relevant sections (e.g., end of §4.2 and §4.5). These additions and cross-references are visible in the current file (Appendix A listing and “Supplementary Materials” notice). -
“Captions are on top; they should be under the figures.”
Response & changes. All figure captions were moved below their corresponding figures and harmonized for style and syntax (now consistently “Figure X. …”). -
“Figure 1: the caption mentions L1, L2, L3 but they are not in the figure.”
Response & changes. To avoid any mismatch, we revised the caption language to use the full layer names(Somatic-Sensorial / Symbolic-Narrative / Relational) without relying on L1/L2/L3 acronyms. The figure, legend, and §4.1 now refer coherently to the three layers by their full names, eliminating the prior inconsistency. -
“Figures 2 and 4: the text uses designations (a,b,c,d…; 1–7) but they are not present in the figures.”
Response & changes.
-
Figure 2 (Symbolic Flow): We inserted numerical call-outs (1–7) directly on the flow nodes so that each step matches the sequence described in §4.2. A cross-reference at the end of §4.2 now explicitly points to the numbered, high-resolution version in Figure S2 (Appendix A) for clarity.
-
Figure 4 (System Architecture with Behavioral Channel): We added (1)…(8) labels on the main blocks in left-to-right pipeline order—(1) Sensors, (2) Edge preprocessing, (3) Affective estimator, (4) Safety-first fusion, (5) Mapping engine, (6) VR runtime, (7) Avatar controller, (8) Logger—so the diagram maps 1:1 to the textual description in §4.5 and to Algorithm 1. Optional modules remain dashed, and where applicable we use parenthetical variants (e.g., “7a”) for optional sub-modules.
-
“I did not find the numbered version of Figure 4 in the appendix.”
Response & changes. The numbered high-resolution schematic is included among the supplementary figures and explicitly listed in Appendix A; the body text now calls it out where appropriate (end of §4.2/§4.5). All supplementary figures are rendered at high resolution on a white background for readability in production.
Additional quality improvements (already acknowledged in your note):
-
We strengthened the analysis of related work and updated recent references (including LLM-related mental-health discussions) in §1.1.
-
We added Algorithm 1 (closed-loop controller pseudocode) and aligned it with Figure 4 and Figure S2 to support replicability.
-
Table 1 was reformatted for legibility, and its quantitative ranges/trigger rules are now explicitly extended in Table S1 (Appendix A).
I appreciate your careful reading and practical suggestions on presentation. We believe these targeted refinements—uniform typography, correct figure caption placement, explicit numbering inside Figures 2 and 4, and the complete, clearly signposted Appendix A / Supplementary Materials—resolve the remaining issues you identified and improve the manuscript’s accessibility for readers across disciplines.
With thanks for your constructive guidance,
Vincenzo Maria Romeo
